


# Flash Flood warning in context: combining local knowledge and large-scale hydro-meteorological patterns

Agathe Bucherie[1,2,3], Micha Werner[2], Marc van den Homberg[3], Simon Tembo[4]

[1] The International Research Institute for Climate and Society (IRI), Columbia University, New York, 10964, USA
[2] IHE Delft Institute for Water Education, Delft, 2611AX, The Netherlands
[3] 510, an initiative of The Netherlands Red Cross, The Hague, 2593 HT, The Netherlands
[4] Malawi Red Cross Society, Lilongwe, 30096, Malawi

*Correspondence to*: Agathe Bucherie (agathe.bucherie@gmail.com)

**Abstract.** The small spatial and temporal scales at which flash floods occur make predicting events challenging, particularly in data-poor environments where high-resolution weather models may not be available. Additionally, the uptake of warnings may be hampered by difficulties in translating the scientific information to the local context and experiences. Here we use social science methods to characterise local knowledge of flash flooding among vulnerable communities along the flat Lake Malawi shoreline in the district of Karonga, northern Malawi. This is then used to guide a scientific analysis of the factors that contribute to flash floods in the area using contemporary global datasets; including geomorphology, soil and land-use characteristics, and hydro-meteorological conditions. Our results show that communities interviewed have detailed knowledge of the impacts and drivers of flash floods (deforestation, sedimentation), early warning signs (changes in clouds, wind direction and rainfall patterns), and distinct hydro-meteorological processes that lead to flash flood events at the beginning and end of the wet season. Our analysis shows that the scientific data corroborates this knowledge, and that combining local and scientific knowledge provides improved understanding of flash flood processes within the local context. We highlight the potential in linking large-scale global datasets with local knowledge to improve the usability of flash flood warnings.

## 1 Introduction

Weather related hazards are responsible for 78% of the economic losses and 38 % of the fatalities related to disasters worldwide, with a drastic increase in the number of events in the last 35 years attributed to global climate change. Hydrological events show the highest increase globally with a rise of a factor of four, while meteorological catastrophes have increased a factor of three (Hoeppe, 2016). Although these events affect the entire globe, exposure to hydrological events and vulnerability of those affected are not uniformly distributed, and climate risk disproportionately affects the



world's poorest (Byers et al., 2018). For example, the impact floods have is greater in developing countries. Indeed, 95% of people are affected by floods, and 73% of the total direct damages every year occur in Asia and Africa (Alfieri, et al., 2017). Aiming at reducing the global impacts of natural hazards, the Sendai Framework for Disaster Risk Reduction (UNISDR, 2015) calls for increased adoption of multi-hazard early warning systems.

A recent survey of the development of operational forecasting systems for floods (Perera et al, 2020) shows that in many
countries and river basins good progress has been made, though such progress is often limited in least developed countries, hampered by a lack of monitoring networks as well as human and technical capacities. These also often focus on large space and time scale riverine floods, which have attracted most attention of the flood forecasting, warning and response research community (Alfieri, et al., 2018, Kauffeldt, et al., 2015, Sai, et al., 2018). Flash floods, in contrast, occur at smaller spatial and temporal scales, resulting in severe damage to infrastructure and the environment, and are more deadly than riverine
floods (Jonkman, 2005). Flash flood events are characterized by very rapid runoff generation and the sudden rise of water levels out of the riverbanks. They can be caused by a combination of high local precipitation rates (Doswell, 1995), adverse antecedent hydrological conditions (Hill and Verjee, 2010) and the geomorphological disposition of the catchment to flash flooding (Georgakakos, 1986).

Flash flood warning is challenging due to the response times of the catchments that flash floods occur in. These are often
shorter than the time needed for decision making, thus preventing efficient flash flood warning responses (Drobot and Parker, 2007). Additionally, the development of effective warnings for flash floods is hampered by the spatial and temporal incoherence between the understanding of the atmospheric and geomorphological processes that leads to flash floods, and observation data availability, even in countries with well-developed hydro-meteorological networks (Creutin and Borga, 2003). Recent decades have, however, seen significant progress in developing warning systems in flash flood prone
catchments (Hapuarachchi, 2011; Braud, 2018), though these rely extensively on the availability of high-resolution quantitative precipitation estimates and forecasts, in particular helped by the availability of radar-based precipitation estimates and nowcasts (Creutin and Borga, 2003; Werner and Cranston, 2009; Javelle et al., 2010).  Such weather radars are practically non-existent in developing countries. Medium to high-resolution numerical weather prediction (NWP) models may, however, be available and are applied in selected cases, such as in the Flash Flood Guidance System for Southern
Africa (Poolman et al., 2014). Flash Flood Guidance relies on geomorphological indicators (Azmeri, et al., 2016, Smith, 2003) of the susceptibility of a catchment to flash floods and triggers.  Several approaches to establish triggers have been developed across the globe; based on forecasting river discharge (Drobot and Parker, 2007), or rainfall thresholds (Alfieri, et al., 2015, Georgakakos, 2005). These also rely on the availability of hydrological observation data for calibration and validation of triggers and have reached a high level of complexity. Availability of these data is not equally distributed around
the world, compounding the difficulty of making flash flood predictions in data poor countries, which are often also developing countries. Global and continental scale flood forecasting systems are increasingly being developed with the availability of global meteorological forecast and reanalysis datasets as well as satellite-based precipitation data (Emerton et al. 2016), and these provide an opportunity to fill the gap where national and regional forecasting systems are not available.


However, although these offer the advantage of providing consistent datasets to areas otherwise poorly served, the limited
resolution of global and continental scale NWP datasets means they are inadequate to support flash flood forecasts (Emerton et al., 2016), and may be limited only to the forecasting of larger scale weather patterns.

Despite these technical challenges, increasing the ability to anticipate the occurrence and impacts of flash floods stands to benefit communities at risk and organisations involved in disaster relief, potentially leading to faster response and better allocation of emergency flood relief effort. However, to be effective, the early warning needs to have not only a technical
basis, but also a human-centred approach, commensurate with the knowledge of the people at risk (Basher, 2006). Local communities have shown to have a complex knowledge cutting across the full disaster risk management cycle (Šakić Trogrlić, et al., 2019) and the climatic conditions that lead to extreme (flood) events (Lefale, 2010; Orlove, 2010). Integration of both local and scientific knowledge is recommended in all steps of early warning system design (Martin, 2012), and can contribute to closing the "usability" gap (Vincent et al. 2020). Plotz et al, 2017 suggest two approaches to
integration of local knowledge and the knowledge derived from contemporary forecasts systems; either through validating local knowledge based on scientific datasets, or through combining the local and scientific data into a consensus forecast that considers both knowledge. The evolving people-centred paradigm to early warning also recognises that community engagement, integration of local perceptions and information tailored to those at risk is important to the fostering of trust in warning information, thus increasing the potential of its uptake (WMO, 2015, 2018). Impact-based forecasting, follows this
paradigm, recognizing that early action by those at risk is more likely to be taken where warning messages recognise people's local understanding of the hazard, environmental and social cues (Calvel et al., 2020), and potential impact (Luther et al., 2017, Meléndez-Landaverde, 2020).

In this paper we explore local knowledge and science-based information on the occurrence of flash floods in rural communities in Karonga District in northern Malawi. We hypothesize that local knowledge can complement the information
contained in larger scale global datasets, and that the combination of local and scientific knowledge can contribute to the development of meaningful and trustworthy early warning. North Malawi is an example of an area with high flash flood risk where the population is extremely vulnerable due to low coping capacities. Through interviews with impacted communities, we develop a joint understanding of the root causes of flash floods in the area, the impacts these have and where these are more likely to occur. In the interviews we also consider the local knowledge of meteorological and hydrological signs
communities recognize as precursors to flash flood events. We then interrogate available information on catchment geomorphology and hydro-meteorological conditions derived from large-scale global models and satellite datasets to examine if these provide useful information congruent to that local knowledge. Our aim is to reconcile these scientific data with local knowledge of flash floods to inform the implementation of people-centred flash flood warnings and foster the taking of early action by communities.



## 2 Study area

Malawi is a Sub-Saharan landlocked country in South-East Africa, sharing its borders with Zambia, Mozambique, and Tanzania. It has an elongated orientation, following Lake Malawi, with its physiography dominated by the rift valley geology. The subtropical climate, and highly seasonal precipitation variability, result in Malawi being prone to weather-related disasters (McSweeney et al., 2010). Two main seasons exist: a wet austral summer season (Nov-April) and a dry season (May-Oct). Perhaps more importantly than its disposition to weather-related hazards, the most severe impacts from disasters result from the high vulnerability of the population, estimated at around 16.5 million, with a poverty Index of 57.9 %. Karonga is the northernmost district of Malawi, sharing a border with Tanzania. It is located along the Lake Malawi shore, has a surface of 3355 km2 and an estimated population of 380,000 in 2020. The district is characterized by a steep rift escarpment separating the hills and plateau area from the lake-shore plain (see Fig. 1). This district, being one of the most vulnerable to floods, is the focus of several projects of the Malawi Red Cross Society that aim to improve preparedness, early action and disaster response in Malawi.

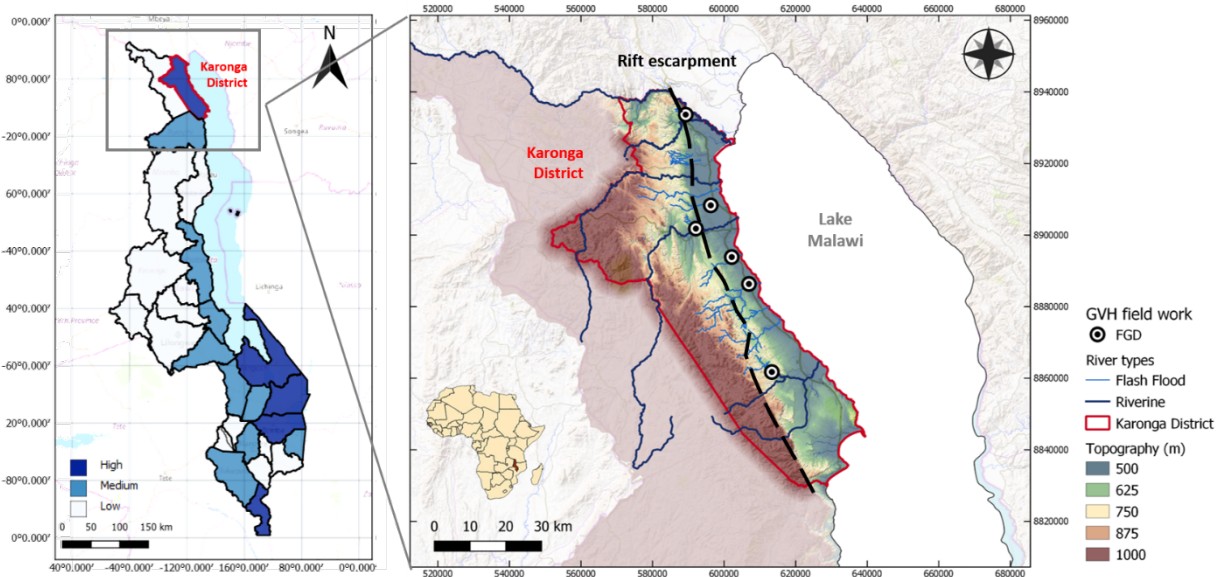

**Figure 1: Karonga district location on - (left) Malawi flood frequency map (ICA 2015) - (right) Topography map showing the rift escarpment, the main rivers, and the six visited villages where Focus Group Discussion were conducted for the research.**

## 3 Materials and methods

### 3.1 Building common knowledge of flash flooding

To develop a common understanding of what constitutes a flash flood in the perception of local communities in Karonga District, a primary data collection campaign was conducted at national, district and community levels through a series of





semi-structured Key Informant Interviews (KII) and Focus Group Discussions (FGD), following a systematic protocol (see
Section 3.1 of supplementary materials). The KIIs at national and district level, and FGDs held at community level followed
a similar questioning pattern allowing, for an alignment and comparison of the results obtained.

KII refers to qualitative in-depth interviews with people from a range of sectors selected for their knowledge on the specific
topic (USAID, 1996). Six KII were carried out in English at the national level; including identified experts from the Malawi
Red Cross Society, the Department of Disaster Risk Management, the Ministry of Finance Economy and Development, the
Department of Climate Change and Meteorological Services, the Ministry of Agriculture Irrigation and Water Development,
and Mzuzu University. At the Karonga district level, local actors such as members of the district civil protection committee,
a reporter from Nyasa Times news agency, and NGOs active in the district (Salvation Army, Focus and Self-Help Africa)
were interviewed. Based on these interviews, we identified twelve communities that are recognised to have a higher flash
flood risk in Karonga District, located on different river basins across the district.

Focus Group Discussions (FGD) were carried out with six of these communities (communities are identified as a Group
Village Head, GVH) represented as circles in Fig. 1. FGD interviews bring together a group of persons from a similar
background guided by a facilitator. We made sure that at least four persons in the FGD were above 50 years old, with at least
one participant having spent more than 50 years in the community in each group. These were conducted and recorded in the
local languages, Nkhonole and Chichewa; and subsequently transcribed and translated into English. A community drawing
exercise was held in each village, resulting in a map gathering information about historical flash flood frequency, impacts,
and perceptions of flash flood risk in different areas of the community. This was followed by a transect walk through the
most affected part of the community. In addition, information about historical flash flood occurrence and their impacts was
collected from each community. From all KII and FGD scripts, local knowledge was extracted and coded into thematic
analysis following five themes; (i) Flash flood definition; (ii) occurrence of flash flood events; (iii) impacts of flash floods
(iv) risk perception, and (v) the signs leading to flash floods. These themes were based on the dimensions of local knowledge
of the flood risk management cycle identified by Sakic Trogrlic et al. (2019), in particular considering the dimensions related
to knowledge of meteorological and riverine indicators, and flood hazard.

We complement these primary data with secondary data on historical flash flood events, their location, date and recorded
impacts. These were extracted manually from five different sources of information: Disaster reports from Humanitarian
Actors (IFRC GO, n.d.; UNICEF Malawi, n.d.), existing global disaster (EM-DAT, n.d.; Munich RE, 2004), government
data, online news-briefs (FloodList, n.d.; ReliefWeb, n.d.) and national online media (Nyasa Times, n.d.; The Nation, n.d.).
All datasets were filtered and consolidated into an event database at three levels of spatial granularity: Karonga District,
Traditional Authority (TA) and community levels (GVH). A total of 142 records of flash flood events affecting the district
from 2000 to 2018 were gathered (Bucherie, 2021). This included 48 events reported at district level, and 38 events at TA
level. Only 18 events are reported to affect the six communities of interest, covering the period 2004-2018.



## 3.2 Mapping flash-flood susceptibility based on scientific data.

The susceptibility of an area to be affected by flash flood depends on the geo-morpho-metric and surface characteristics (Horton 1945, Patton, 1976), which have a strong influence on catchment hydrologic response to heavy rains, and therefore on runoff generation. Here we map the relative susceptibility to flash flooding of the twelve communities identified to have

the highest flash-flood risk in Karonga District. For each of the 12 communities, hydrological catchments are delineated, using the global SRTM 90m Digital Elevation Model (DEM) v4.1 (Jarvis et al., 2008). Geomorphological indicators related to surface and morphometric characteristics known to characterize flash flood risk are identified (Bajabaa, et al., 2013, Farhan, et al., 2016, Rogelis and Werner, 2014), with some linked to the local knowledge (such as indicators related to slope, soil type, land or vegetation cover). The identified indicators and references are described in Table 1.

For each catchment, geomorphological indicators are calculated and classified according to four categories characterizing the geometry, the hypsometry, the drainage network, and surface of the catchments (Appendix A). While the first three categories of catchment indicators are extracted from the DEM analysis only, the surface characteristics indicators are derived from the Malawi government soil type and Land-use Land-cover (LULC) data, and the Normalized Difference Vegetation Index (NDVI) Copernicus Global Land Service 300m product (Roujean et al., 2018).  All geomorphological

indicators are normalised from 0 to 1 according to their contribution to susceptibility to flash flooding.

Different methods of weighting the influence of each geomorphological parameter can be used depending on the context and the scale of each case study (Azmeri, et al., 2016). Some studies use equal weighting (Zogg & Deitsch, 2013), or weighting based on the indicator ranking (Karmokar & De, 2020). Here we apply a weighted method based on Principal Component Analysis (PCA) to reduce the dimensions in each class (Chao & Wu, 2017, Rogelis & Werner, 2014). Based

on the four Principal Component results, a ranking of flash flood susceptibility of the 12 catchments is calculated, following Eq. (1), representing the inherent potential of each catchment to generate a flash flood in case of heavy rain. PC[name] refers to principal components related to geometry, hypsometry, surface and drainage network. The linear coefficients a,b,c and d are the weights applied to each of these classes.

$$FF_{Suscept} = a \times PC[geom] + b \times PC[hypsom] + c \times PC[drain] + d \times PC[surf] \qquad (1)$$





**Table 1. Identified geomorphological and surface indicators related to flash flood susceptibility**

| | Index | Description | Source |
|---|---|---|---|
| **Catchment geometry** | Area [km²] | The area of a catchment is correlated with its discharge. Bigger catchments have lower flash flood potential. | *Gray (1961)* |
| | Length to width LtoW [-] | Length to Width ratio is a shape indicator of a catchment, inversely proportional to the flash flood potential of a catchment. | *Schumm (1956)* |
| | Basin circularity Bc [-] | The circularity ratio $4*\Pi*A/P^2$ is related to flow peak and debris flow occurrence. The more circular the basin, the more flash flood prone it is considered to be. | *Miller (1953)* |
| | Time of concentration. Tc [min] | Flash flood potential increases in basins with lower time of concentration. Computed using Kirpich's formula, where L is the longest flow path to the remotest point altitude H from the outlet at elevation h. $$tc = 0.01947 \times L^{0.77} \times \left(\frac{H-h}{L}\right)^{-0.385}$$ | *Kirpich (1940)* |
| **Hypsometry** | Average slope [deg] | The average slope is an indication of the flashiness of a watershed, proportional to flood susceptibility. The slope is computed in degrees from the DEM using 2nd degree Polynomial Adjustment algorithm (Zevenbergen & Thorne, 1987) | *Lindsey et al. (1982),* |
| | Rel_Relief [-] | The Relief (H) is the difference between the highest and the lowest elevation point of the catchment. The Relative Relief is the Relief (H) divided with the basin perimeter (P). It is related to the event magnitude. | *Melton (1957)* |
| | Elv_RR [-] | The Elevation Relief Ratio definition using the Relief (H) divided by the Length of the catchment has been used. | *Oruonye (2016)* |
| **Drainage network** | Drainage relief ratio D_RR [-] | The Drainage Relief ratio corresponds to the drainage relief (maximum elevation of the drainage system minus the outlet elevation) divided by the longest stream length. | *Schumm (1956)* |
| | Drainage density Dd [Km⁻¹] | The cumulative river length over the area. This has a direct correlation with flood potential. | *Patton and Baker (1976)* |
| | Bifurcation ratio Rb [-] | The bifurcation ratio of a basin is inversely linked to the flash flood risk: Rb = ∑ Nu/Nu + 1, where Nu is the total no. of stream segments of order "u", and Nu +1 the no. of segments of the next higher order. | *Strahler (1957); Schumm (1956)* |
| **Surface properties** | Soil type | Soil types data are ranked according to the clay content, and reclassified from 0 to 10. The infiltration rate decreases with the clay content, increasing runoff and flash flood susceptibility. | *Smith (2003); Tincu et al. (2018)* |
| | LULC | Land Use Land Cover data are reclassified into classes from 0 to 10 depending on their susceptibility to increased flash flood risk. | *Smith (2003); Tincu et al. (2018)* |
| | NDVI | The Normalized Difference Vegetation Index is used as an indicator of the greenness of the vegetation. Data are extracted for two periods, at the beginning (21 to 31 Dec 2017), and at the end (21 to 31 March 2018) of the wet season. High NDVI values represent dense vegetation while low NDVI values indicate bare soils and increasing flash flood risk. | *Smith (2003); Tincu et al. (2018)* |


Validating a map of flash flood susceptibility is a challenge where there is little historical data (Alam et al., 2020). For data rich catchments, machine learning techniques use historical flash flood data to calibrate the flash flood susceptibility map (integrating morphometric and precipitation indicators) and have been tested on specific catchment scale study (Arabameri et al., 2020; Pham et al., 2020). Here we estimate the weight (a, b, c, d) of each class by calibrating against the estimated

relative flash flood frequency for each catchment as indicated by the communities interviewed. The best fit is defined





minimizing the Root Mean Square Error (RMSE) between the modelled susceptibility ($FF_{Suscept}$) and the normalised observations of flash flood frequency.

**3.2 Identifying hydro-meteorological conditions associated with flash flooding.**

Precipitation and large-scale hydro-meteorological indicator datasets are selected, guided by the knowledge gained from
communities on the signs and triggers they consider as precursors to flash floods. The spatial and seasonal distribution of indicators derived from the datasets are analysed to understand if these corroborate with the reported signs, and particularly if these reflect conditions associated with flash floods during and prior to the catalogued historical flash flood events.

Precipitation is derived from the GSMaP satellite-based precipitation products (Aonashi, 2009, Okamoto, 2005, Kubota, 2007), and extracted for the 15 wet seasons of the 2002-2018 period. We limit the extraction of these data to the wet season
for computational reasons, as well as due to flash flood events occurring in the wet season. GSMaP was selected given the high spatial (0.05 degrees) and temporal (hourly) resolution, as well as relatively low latency. Historical extreme rainfall patterns are explored spatially and temporally over the district. In addition, maximum daily 1-hour and 3-hour rainfall totals are extracted to characterise precipitation intensity associated with the 18 catalogued flash flood events affecting the six communities for the 2004-2018 period. Moving windows of 6h, 1-day and 3-day cumulative rainfall are extracted as
indicators of antecedent cumulative precipitation. These precipitation indicators and associate statistics are extracted from the GSMaP data at locations corresponding to the centroids of the catchments of interest. Time series are analysed visually for each flash flood event.

Large-scale hydro-meteorological conditions are derived from the ECMWF ERA5 climate reanalysis dataset (Hersbach et al., 2020) provided through the Copernicus Climate Change Service (C3S, 2017). This dataset is selected given its
availability and as it provides the same parameters and at similar temporal and spatial scale as the forcing data used in global hydro-meteorological forecast models such as GLOFAS (Alfieri et al., 2013). Daily data is extracted for the 2000-2018 period from ERA5 over a geographical box that encompasses the study area (longitude 32 to 36, latitude -8 to -12). Variables considered include the 2m surface air temperature; the 2m dew point temperature; the volumetric soil water content of the first 7cm of the land-surface; the relative humidity of the deep troposphere; the CAPE indicator (surface based convective
available potential energy), and the surface u and v wind vectors.

These variables are extracted from the ERA5 data at three locations (see Fig. 2), one in the northern part of the district (N), one in the southern part (S) and one in Lake Malawi (L). Daily averages of the selected ERA5 variables are extracted for the period 2000-2018 to analyse seasonal variations. To study the larger scale hydro-meteorological patterns and conditions associated with historical flash flood events, the same ERA5 hydro-meteorological variables are extracted and averaged over
four regions (see Fig. 2), for up to 3 days before each of the catalogued flash flood events. These regions are selected based on the precursor signs reported by communities interviewed, and include Karonga (Region W), Lake Malawi (Region S), and the areas to the north-west and north-east of Karonga (Region NW and NE respectively).


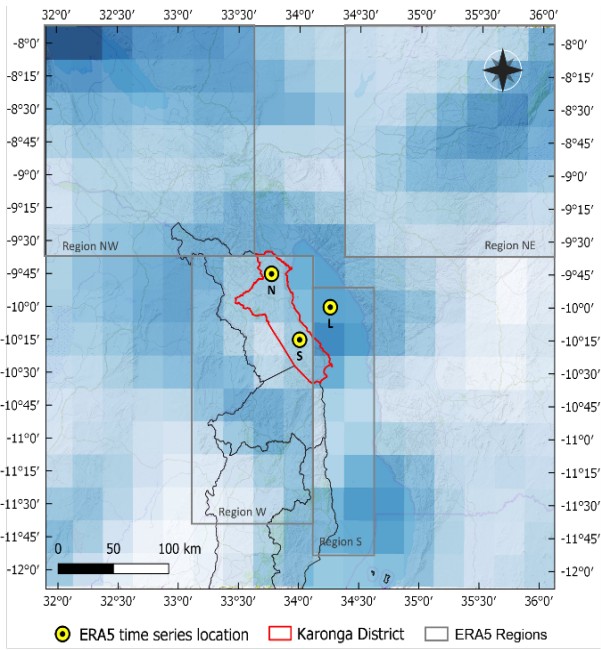

**Figure 2: ERA-5 grid resolution, x,y point locations (yellow dots) and four regions (NW = North West, NE = North East, W =**
**West, S = South) used for large-scale hydro-meteorological pattern analysis.**

**4 Results**

**4.1 Building knowledge on Flash flooding.**

Local knowledge on flash flooding in Karonga district has been compiled from the thematic analysis of the local knowledge
(extracted from all KII, FGD, transect walks and community drawing), and integrated and corroborated with the flash flood
occurrence and impact analysis from the secondary data. The resulting compilation is structured using the five themes
identified in the collection and analysis of primary data.

*(i) Flash flood description*

Communities explain that they experience sudden floods, induced by intense and short rainfall events, and that these form
the main type of weather induced disasters in Northern Malawi. All communities describe these flood events as unexpected
and occurring "*without notice*". These floods are characterised by an intense power of river flows, eroding of river banks and
rivers bursting their banks. The topography is known to govern the occurrence of flash floods in Karonga, and the most
affected areas are on the flat rift valley floor close to the escarpment. In addition, fast onset floods sometimes coincide with
slow onset riverine floods in communities in the North of Karonga; a scenario described as resulting in larger scale, longer
duration, and more severe disasters.



*(ii) Occurrence of flash flood events*

The analysis of the data from the KII and FGD, supported by the analysis of the secondary data on the spatial and temporal occurrence of flash floods, reveals that flash floods happen between one and eight times per year in Karonga district, mostly

in January and in March/April, and generally overnight. In addition, shorter duration and more localised flash flood events are reported to occur in January, while longer duration floods affecting larger areas are observed in April (Table 2). The frequency of flash floods is found to be higher in the northern part of the District, in Kyungu and Kilipula TA (see Fig. 3). In addition, April events affect mainly catchments in the northern TAs while January events may affect any of the catchments in the entire district.

**Table 2: Monthly flood event frequency based on 2000-2018 secondary data collection (43 recorded events), and associated proportion of short duration (<=3days) and local (affecting only 1 TA) recorded flood events.**

|  | December | January | February | March | April |
|---|---|---|---|---|---|
| **No. of recorded events** | 6 | 15 | 4 | 7 | 11 |
| **% short duration flood (<=3days)** | 83 | 80 | 100 | 86 | 55 |
| **% local event (affecting only 1 TA)** | 66.6 | 61.5 | 50 | 28.5 | 33.3 |

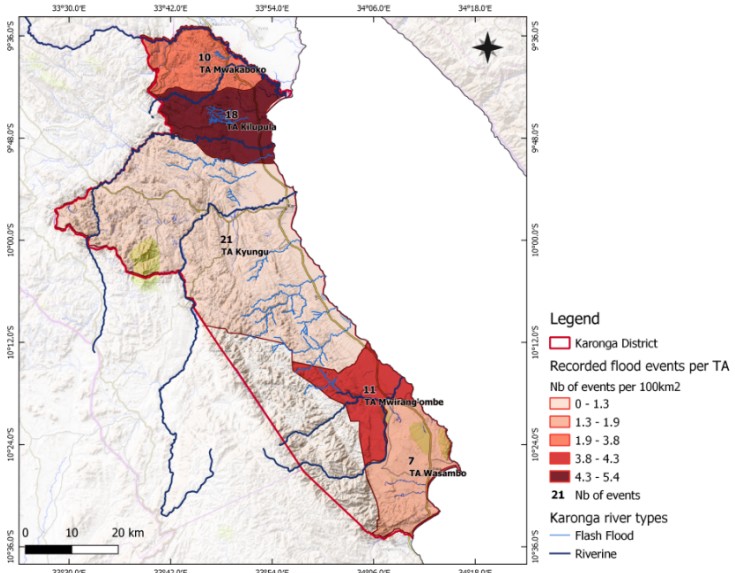

**Figure 3: Number of recorded flood events per 100 km² for every TA (collected for the period 2000-2018), with the total numbers of events recorded for each TA indicated as labels.**




*(iii) Impacts of flash floods*

The main impact of flash floods experienced in Karonga is on agriculture, as flash flood events are reported to systematically
sweep away several hectares (ha) of crops (either recently planted or fully grown), and sometimes livestock. In addition, communities mentioned that flash floods can destroy parts of villages, irrespective of the type of buildings. Communities report an increase in flood impact since 2007. Based on our historical data, we estimate that when a flood occurs in the district it has a 50% chance to affect at least 300 households, 200 ha of crops and kill at least one person. The communities also indicated that the impact of floods is higher in April, and also higher in the North of Karonga. This is attributed to the
combination of flash floods with riverine floods in the Songwe, Kyungu, Lufira and North Rukuru rivers, as well as the higher population density in the North. These prolonged and larger-scale flood events are known to trigger cholera outbreaks in the region. Flash flood events occurring in January can, however, affect the entire district. Their impacts on communities are more spatially contained, but maybe more severe at that small scale.

*(iv)Perceived drivers of flash flood risk*

A summary of six factors perceived to increase flash flood risks in Karonga District is presented below, including quotes where relevant made by different community members during the FGDs.

**River sediments**: Karonga has mostly ephemeral rivers with heavily silted river beds, which are dry from the months of August to December. Increasing sedimentation of the river beds and changes in river behaviour are reported by elderly
people. High sedimentation in rivers is recognised to have increased flood occurrence, and the exposure of people living along these rivers. *"Before, water would swell within its course and go back to normal without any damage. Today, rivers are full of sand, blocking culverts, and preventing water from flowing in its original course"*. River profiles where these cross the lake-shore plain are very flat, and their channels were observed during our transect walks to not be deep enough to accommodate peak discharges.

**Land use**: The sedimentation in the river courses is identified by all communities to be the result of deforestation in the upstream catchments, which started in the 1970's, after the independence of Malawi. Charcoal production, and the use of wood to build stronger burnt brick structures have risen; *"Bushfires and overgrazing animals are leaving the soil bare and prone to flood"*. In addition, the intensification of agriculture along the river banks is known to reduce the natural control of the water velocity, increasing the spread of water in the farmland.

**Climate change:** A shift toward shorter and more intense rainy seasons (December to March instead of November to April) is observed by communities, leading to more frequent and devastating events. An intensification of rain events and a change in meteorological patterns is described by elderly people, making the indigenous prediction more difficult than before.

**Geomorphology**: Fast running water is experienced to come from the steep slopes of the mountains of the escarpment, and affects villages in the low land areas, making the proximity to the hills a factor of increasing risk.

**Soil type:** The relation between flash floods and soil type is expressed in terms of erodibility of the soil; *"The soil, easily swept away during flood, is dispersed in agricultural fields and lowers soil fertility"*. In the Northern communities, a relation



between the flood duration and the clay content in the soil, lowering the infiltration capacity in the lake margin plains, is observed.

**Socio-economic vulnerability:** The rapid increase of population and poverty, associated with poor settlement and farming
practices, has exacerbated the vulnerability of communities in the district.

*(v) the signs leading to flash floods.*

The local knowledge of the signs experienced prior to flash floods are categorised into two types of observations.

**Meteorological signs:** Community knowledge revealed that flash floods are associated with strong south-easterly winds at
the end of the wet season. In addition, communities in the South of Karonga District described flash floods to be associated with highly localised storm events and thunder, black and slow-moving convective clouds, and changeable wind direction. In addition, a rise in temperature before flash floods was reported by all communities.

**Hydrological signs:** When water velocity and volume increase in the rivers, the amount of debris carried as well as the colour of the water changes (turning brown, black, milky or red), announcing that a major flood is coming. In addition, at the
lake margin, the soil moisture and water table are known to be high at the end of the wet season, increasing the duration of floods and consequent impacts. When the soil is dry, water from the flash floods will infiltrate faster in the farmlands.

## 4.2  Geomorphological analysis of flash-flood susceptibility in Karonga District

All catchments that drain the escarpment are reported to be susceptible to flash flooding in Karonga district although the level of susceptibility may vary between catchments. The results from the flash flood frequency estimation from each
community was fundamental to understanding the spatial difference in flash flood susceptibility from the geomorphology and surface characteristics of the upstream catchments upstream. The normalised indicator values and PCA composite results calculated for each catchment are presented in supplementary material section 4.2; the PCA component loadings are found in Appendix A.

The relative catchment susceptibility results for the four geomorphological classes; geometry, hypsometry, drainage system
and surface characteristics; reveals differences from North to South, as shown in Fig. 4. The dashed coloured lines show the results from the PCA analysis for each of the four categories of catchment characteristics. Black triangles show the normalised frequency of flash floods as reported by the communities visited, which were used to estimate the weights (a,b, c and d) for the final flash flood susceptibility ranking of the selected catchments. The thick solid line shows the normalised flash flood susceptibility indicators, found from the weighted contributing factors; geometry (0.5), hypsometry (0.1),
drainage (0.2), and surface (0.2), where the values in brackets are the weights in Equation 1 with the best fit (RMSE of 0.31). The resulting indices are mapped in Fig. 5.

The increased flash flood susceptibility of catchments appears to be mostly driven by catchment geometry and is inversely proportional to the area and the time of concentration (Tc) of the catchment, together explaining 63% of the variance of the geometric class. While the high frequency of flash floods experienced by the community of Iponga and Nkhomi are


explained mainly by their small upstream catchments size (lower than 10 km2) and Tc of about 40 minutes, the lowest flash flood frequency observed in Sabi community can be explained by the largest contributing Area (335 km2) and associated Tc (~4 hours).

The derived weights attribute an equally important influence to both the drainage characteristics (controlled essentially by the drainage relief ratio) and the surface characteristics. While the higher flash flood susceptibility of Iponga and Nkhomi

catchments is also explained by the high drainage relief ratio, spatial variation of the susceptibility related to surface characteristics is mainly driven by the soil types and the vegetation cover. A strong variability in the NDVI is observed between the North and the South, particularly at the beginning of the wet season, exposing the South (with more bare soils) to higher susceptibility to flash floods. The high susceptibility of Iponga catchment is additionally explained by the presence of clayey soils in the North, decreasing the infiltration capacity. Finally, catchment hypsometry results do not correlate well

with field observations. The resulting low relative flash flood susceptibility of the Kyungu, Kibwe and Kasantha catchments in the North of Karonga, which were not visited, confirms these being ranked as less dangerous by Karonga experts through the KII.

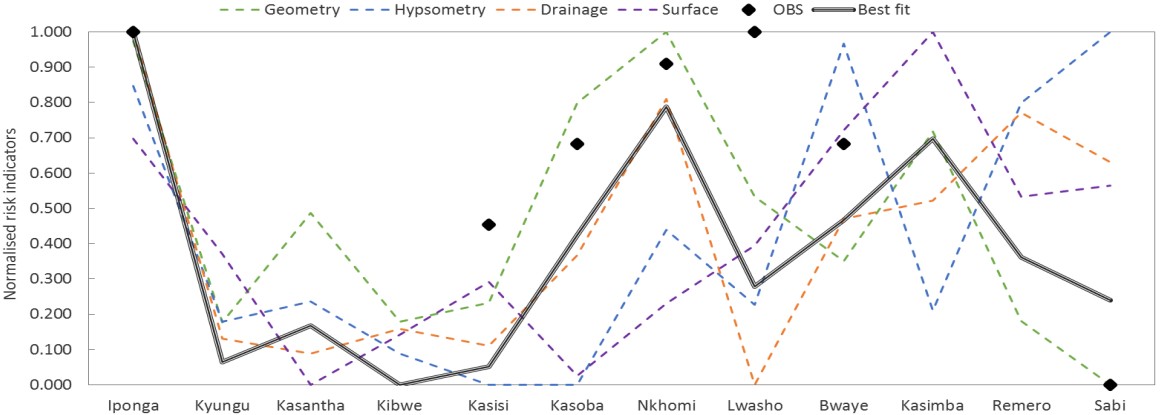

**Figure 4: Results from the PCA analysis of the four catchment characteristic categories and weighted flash flood catchment**
**susceptibility. The black dots correspond to the normalised estimation of flash flood frequency of communities visited (OBS=observed). Catchments are ordered from North to South (left to right).**



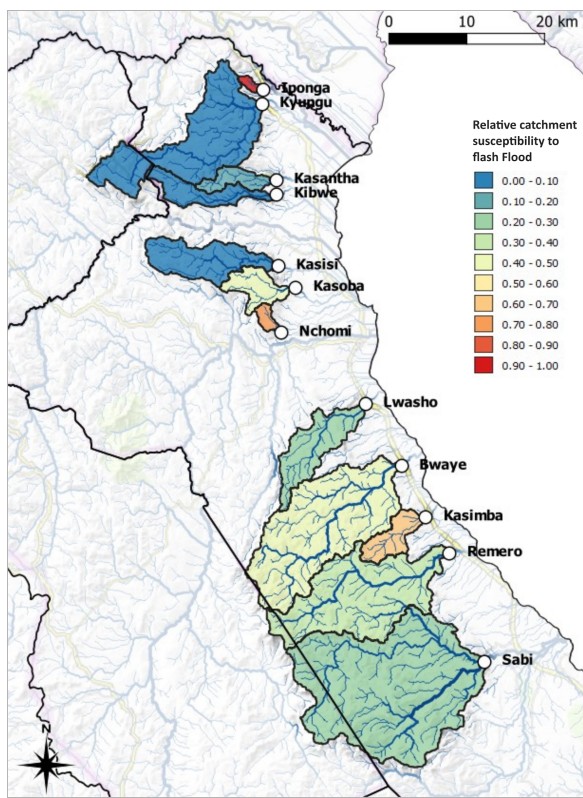

**Figure 5. Relative flash flood susceptibility of the catchments in Karonga district.**

### 4.3 Hydro-meteorological conditions associated to flash flooding

Guided by local knowledge on the hydro-meteorological signs associated with flash flood events, and the temporal distribution of flash flood events through the wet season, we analysed the characteristics of extreme rainfall and large-scale spatial and intra-seasonal hydro-meteorological patterns that could lead to flash floods in Karonga.

The analysis of historical precipitation indicates that heavy rainfall events are not homogeneously distributed spatially, and are distinctly different at the beginning and at the end of the wet season.

Figure 6 shows the maximum daily precipitation found with the GSMaP data, averaged over the time period 2002-2018 for the months of January and April, the months in which flash floods are reported by communities to occur. This reveals that extreme rainfall rates are constrained to the Northern part of Karonga at the end of the wet season in April, while these are distributed more homogeneously in January. A more detailed analysis of the hourly rainfall rate reveals that extreme rainfall events are more frequent in January than in April, though the maximum hourly precipitation rates are comparable in both months (see supplementary material section 4.3.1). All rainfall events associated with the 18 historical flash flood events were detected by GSMaP. It confirms that extreme rainfall peaks of at least 30 mm/h are associated with events localised in the North of Karonga, either at the beginning or at the end of the wet season. In addition, flash flood events that correspond

to low GSMaP rainfall signals (peak rain below 10mm/h) are observed mainly in the South of Karonga, in January and February (see supplementary material section 4.3.2).

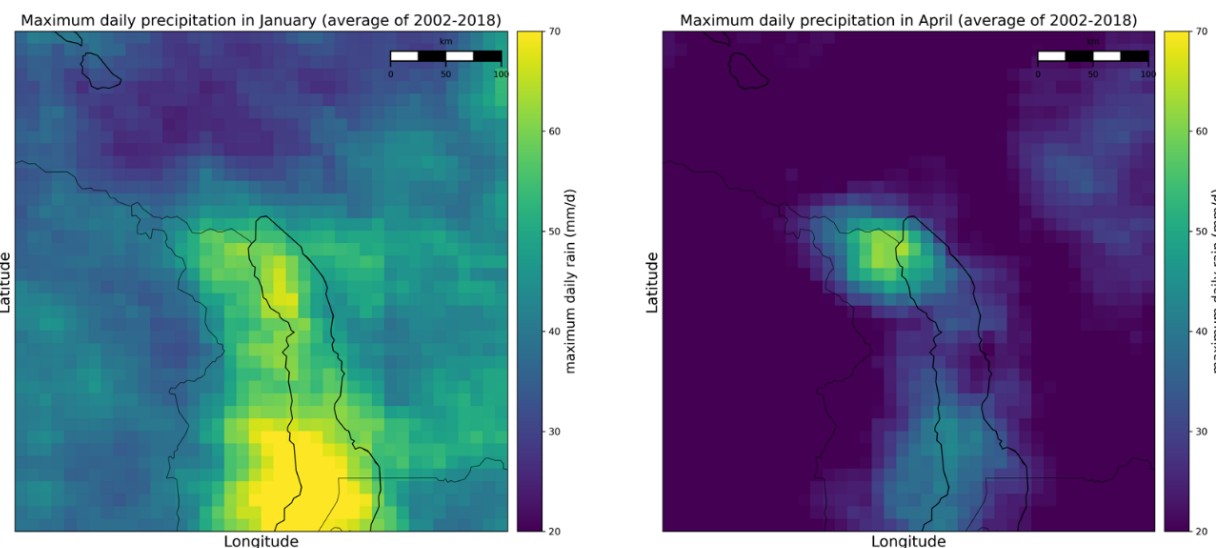

**Figure 6: Maximum daily rainfall over two different months of the wet season (January and April), averaged for 15 wet seasons (2002/2003 to 2017/2018). Yellow colours indicate areas with higher maximum daily rainfall (mm/d) recorded on average for the**
**month. The maps are centred on North Malawi, black lines represent the boundaries between countries and Lake Malawi.**

The large-scale hydro-meteorological analysis on seasonal patterns and conditions associated with flash floods in Northern Malawi help to understand the extreme rainfall patterns in Northern Karonga.

Figure 7 presents the standard daily averages for the selected variables spanning the 2000-2018 period, derived from ERA5
hourly re-analysis and sampled at three locations (Fig. 2).

**The Relative humidity,** which provides an indication of the water saturation of the deep troposphere, strongly increases during the first part of the wet season, from mid-November to end of December. The average relative humidity is at its maximum from January to mid-February, approaching 80%, and is slightly lower from mid-February to the end of March. The relative humidity of the troposphere then drops significantly in April.

**The Soil Moisture** is generally lower in the South than in the North of Karonga. It gradually rises from November to February, then drops at the end of the wet season in the South while remaining high in the North. These results confirm the precipitation observations, showing a prolonged rain season in April in the North of Karonga, but could also be attributed to differences in soil characteristics and vegetation.



**The convective available potential energy** (CAPE) is highly variable within the wet season. It shows a rising trend at the

beginning of the wet season and a falling trend at the end of the wet season, with averaged maximum standard values of 1000 J/km at the end of January revealing a maximum atmospheric instability. In addition, the daily variability of CAPE is higher in January.

**The wind is** generally stronger before December, becoming weaker at the beginning of the wet season. It reaches its minimum intensity in January and February, and increases again towards the end of the wet season. We observe that winds

are stronger over the lake and in the South than in the North of Karonga at the end of the wet season.

During November-December and March-April, the wind is uniformly South-Easterly over land, and Southerly over the lake. During the months of January and February, wind direction is more erratic, characterised by an alternation between North-westerly and North-easterly winds over Karonga and respectively North-westerlies and Southerlies over Lake Malawi.


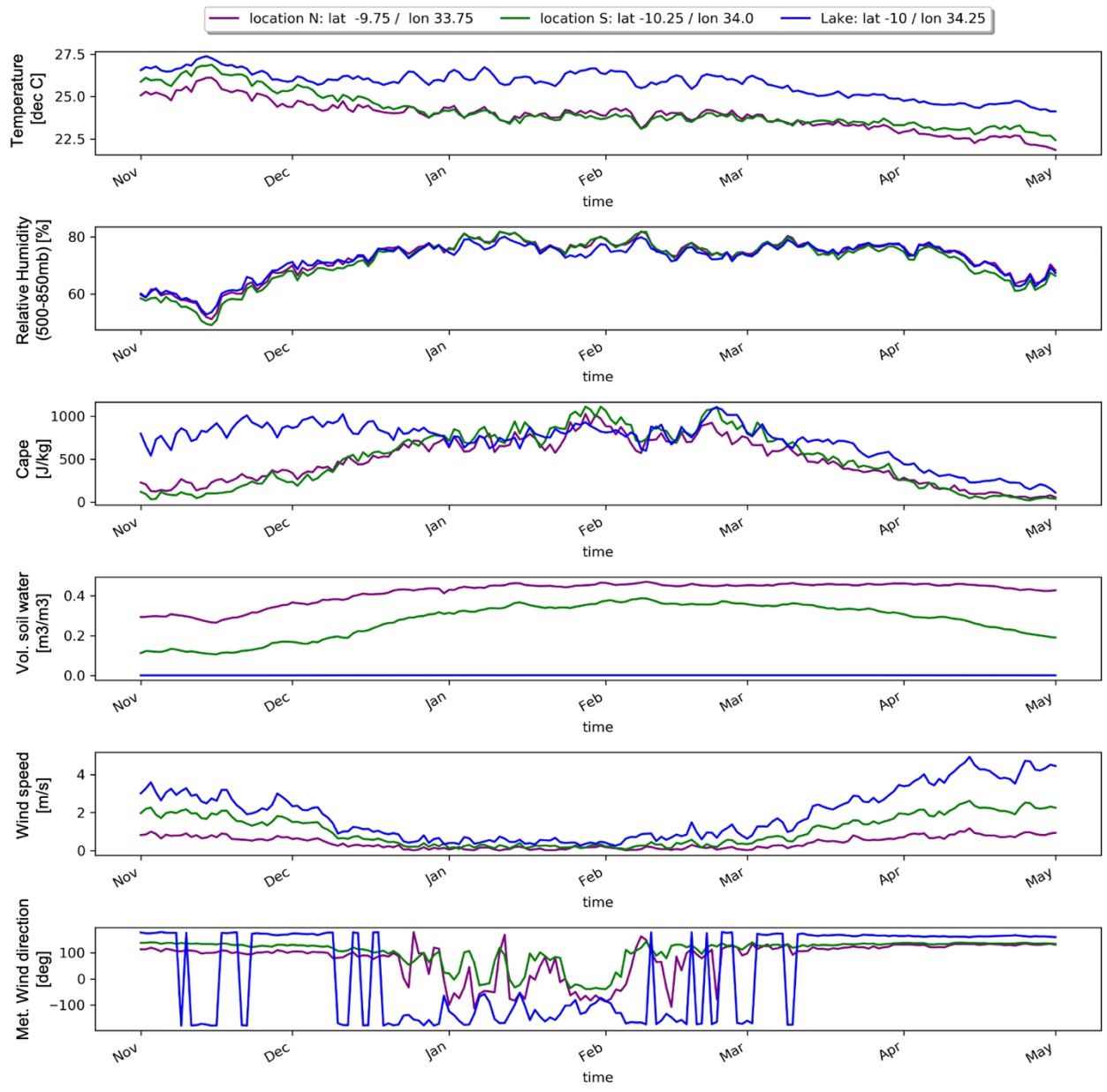

**Figure 7: ERA5 standard daily averaged variables over the period 2000-2018 for the three locations as introduced in Figure 3 representing the North and the South of Karonga, and Lake Malawi. Six hydro-meteorological variables are shown: the surface**





**temperature, the relative humidity of the troposphere, the CAPE factor, the volumetric soil water, the wind speed and the**
**meteorological wind direction.**

Our analysis of these variables confirms our hypothesis of two large-scale hydro-meteorological patterns in northern Malawi, characteristic of the early and late wet season respectively.

The early wet season is characterized by a maximum atmospheric instability in January, with high temperatures and relative humidity. This is when the Inter-Tropical Convergence Zone (ITCZ) is positioned over Malawi. This suggests that the
convective storm risk associated with the tropical climate regime is higher at the beginning of the wet season. During this period, extreme rainfall events are more frequent, more localised, and of shorter duration. Such convective events occur evenly distributed over the district. The wind, alternating between two different regimes, can lead to either orographically enhanced rainfall in the North, or more scattered convective conditions in the South.

The late wet season is driven by an extra-tropical climate regime associated with Mozambique currents coming from the
Indian Ocean. This is consistent with the strong winds from the South observed along Lake Malawi. When this pattern forms, clouds converge toward the northern part of Karonga, where rainfall is orographically enhanced. The North of Karonga experiences a longer rainy season as a consequence, with frequent, intense and continuous rainfall until the end of April, while flash flood risk is considerably reduced in April in the South of Karonga.

While orographic effects in Karonga have been documented (Nicholson, et al., 2014), the differences between the orographic
events of the late wet season and the predominantly convective events of the early wet season as a trigger for flash floods have not been previously studied. These distinct patterns do, however, corroborate the differences observed by local communities as reported in the FGD held.

**4.5 Linking Local and Scientific Knowledge on Flash Flooding in Karonga District**

The local knowledge of the communities in Karonga district on the conditions that lead to flash floods, which we obtained
through the FGD and supported by the KII, was used to guide the diagnosis of scientific data in the exploration of the factors contributing to flash-flood risk in Northern Malawi. In this section we synthesize this local knowledge, and explain how the scientific diagnosis corroborates local knowledge. Table 3 shows the results for the three main themes of analysis (left column), linking the local to scientific knowledge (middle and right columns respectively).





**Table 3: Synthesis of the local and scientific knowledge on flash flooding in northern Malawi**

| | **Local Knowledge** | **Scientific Knowledge** |
|---|---|---|
| **Spatial-temporal occurrence of flash floods** | | |
| Spatial occurrence and annual frequency | Communities experience differences in FF frequency in the different catchments in Karonga, from one event every 3 years to 3 events per year. | Geomorphological features, surface characteristics and precipitation patterns explain spatial differences in FF susceptibility and impacts. |
| Seasonal trend | FF are experienced mostly in March/April, the months receiving the most rain, and sometimes January, related to the first precipitation events of the wet season. | The intra-seasonal hydro-meteorological analysis reveals different hydro-meteorological conditions between the start (January) and the end of the wet season (April), potentially leading to two types of flash flood conditions. |
| Diurnal trend | FF are observed mostly at night | Historical GSMaP precipitation events associated with flash flood events mainly occur during the evenings from 6-9 pm until the early morning 2-4 am in Malawi local time. In addition, a diurnal hydro-meteorological cycle is observed between the land and the lake. |
| **Geomorphology and surface characteristics** | | |
| Soils | Communities describe clayey soils as an aggravating factor, increasing flood duration, while they experience that sandy erodible soils have negative impacts when transported on crops and grass fields. | Karonga soil type analysis reveals a large fraction of sandy soils in the district. More clayey soils are present in the North, potentially linking up with the longer flood durations reported in the North. |
| Vegetation | All communities see vegetation degradation in the Upstream catchment forest as a factor that increases flash flood risk. | NDVI analysis shows a lower vegetation greenness in the South of Karonga at the beginning of the wet season, exposing the South of Karonga to higher flash flood risks. This is either due to a more intense deforestation rate in the South, more visible during the dry season, or reflecting the natural variability of vegetation between the North and the South of Karonga. |
| Catchment geometry | Communities relate the proximity of the escarpment and hills from their village, with increased flash flood risk. | The analysis of catchment time of concentration (40 min for small catchment to 4 hours for bigger catchments), an indicator of catchment geometry, shows the highest correlation with local spatial observation on flash flood frequency. |
| **Hydro-meteorological Knowledge** | | |
| Precipitation | Short and intense precipitation events are indicated as the main trigger of flash floods for all communities. | The analysis of high-resolution GSMaP precipitation during flash flood events confirms that daily maximum hourly rainfall rates are the most important indicator explaining historical flash flood events. |
| Temperature | An increase of temperature is experienced before flash flood events | Daily Temperatures from ERA5 do not reveal any specific increase before FF events. However, a rise in humidity is observed in ERA5 data before flash flood events, potentially linked to an increase in ambient temperature. |
| Soil Moisture | The higher soil water saturation in the flat plain along Lake Malawi in April is responsible for an increased flood duration. | ERA5 volumetric soil water data confirm higher values during the late wet season and in the North. |
| Wind | Change in wind direction and strength associated with flash flooding. Some communities reported strong winds from the lake as a precondition to flash floods. | ERA5 wind data reveals two different regimes at the beginning and at the end of the wet season, with higher instability during the early wet season potentially linked to LK observation. |
| Storms | Localised storms, with rotating black clouds and thunder are described as conditions associated with flash floods. | ERA5 CAPE and Relative Humidity are good indicators of the susceptibility of convective events developing. These show promising signals of FF potential during the early wet season. |



## 5 Discussion

### 5.1. Validating local knowledge in prediction of flash floods

Karonga district, and more generally most countries in Southern Africa, lacks the availability of high-resolution quantitative precipitation forecasts and high-resolution hydrological models that provide plausible prediction of flash floods (Hapuarachchi, 2011; Braud, 2018). Global and continental scale flood forecasting systems (Emerton et al. 2016, Alfieri et al., 2018) potentially fill this gap, but the current meteorological and hydrological models these use are too coarse to provide reliable hydrological predictions of flash floods at the scale of catchments susceptible to flash flooding (Emerton et al. 2016,

Gründemann et al., 2018), or there is insufficient in situ data to correct bias in forecasts derived from such global systems (Bischiniotis et al., 2018, Lavers et al. 2019). Despite this, our results show that larger scale patterns that are identified to be linked to the occurrence of flash floods in Karonga district based on local knowledge, can be discerned in the coarser global scale models and remote sensing datasets. This highlights the opportunity of local knowledge in helping bridge the temporal and spatial scale gap (Plotz et al., 2017) and in deriving flash flood warnings by interpreting forecasts of larger scales

patterns associated with flash floods in the district using indicators that reflect local knowledge. Plotz et al (2017) propose two approaches to combining local and scientific knowledge in forecasts; a consensus forecast approach and a science integration approach that validates the accuracy of forecasts based on local knowledge using scientific data. Following the second approach, we extract the identified indicators of the critical hydro-meteorological conditions associated with the 18 flash flood events recorded in Karonga and use a simple model to test the predictability of the binary occurrence of flash

floods in the Karonga district. The two indicators derived from ERA5 found to be the best predictors of conditions that may lead to a flash flood are the maximum hourly peak in the CAPE in the three days before an event, and the maximum hourly relative humidity of the troposphere one day before the flash flood. The latter indicator was found to be a good predictor in the early wet season only, confirming that in this period flash floods are primarily induced by convective storms. Conversely, antecedent rainfall conditions are found to have more predictive potential during the late wet season, particularly

for the catchments in the North. We explore the predictability of the binary occurrence of the observed flash flood events with these simple indicators at three spatial scales; i) at the scale of predicting the flash flood events in each catchment; ii) at the scale of predicting a flash occurring in the North and or in the South of Karonga district; and iii) at the scale of predicting the occurrence of flash flood events in the district as a whole. Clearly the sample size is small, particularly for predictions of flash floods occurring in individual catchments. Our results show there is little skill in the prediction of flash floods at the

scale of the individual catchments, as the reasonably high probability of detection (POD) is complemented with high probabilities of false detection (POFD). However, skill improved markedly in predicting the binary occurrence of flash floods when pooling warnings for either the northern or southern catchments, and further still at the scale of the district as a whole (see supplementary material section 5.1). These results underscore the limitation of coarser global datasets in predicting flash flood events, but equally highlights the potential these have. Predictions of the likely occurrence of a flash

flood event, either differentiated to the North or South of the district, or in the district as a whole, could be translated to flash



flood guidance in the individual catchments based on the knowledge of the communities of the relative susceptibility of each of the catchments in the district, and predicted large-scale meteorological conditions. This approach is in principle similar to differentiated rainfall thresholds derived to support flash flood guidance statements such as used in the Southern Africa Region Flash Flood Guidance System (SARFFG) developed in collaboration with the Malawi Department of Climate

Change and Meteorological Services (DCCMS) (Jubach and Tokar, 2016), but differs as it differentiates catchments based on local knowledge, corroborated by the scientific assessment of catchment flash-flood susceptibility. We argue that this contributes to more effective dissemination of guidance on the potential occurrence of flash floods as it considers the knowledge and perceptions of the recipients.

Although our sample size of 14 KIIs and 6 guided FGDs with 7 to 11 persons allowed us to capture the diversity of local

knowledge in the area, we cannot say data saturation was fully reached. Most likely a larger sample size of FGDs would have allowed for a further in-depth spatial characterization of local knowledge, and shed light on minor discrepancies such as why communities considered the Kasisi catchment to be more susceptible than the Sabi catchment, despite the contrary being suggested by the geomorphological characteristics. Nevertheless, the validation of the local knowledge obtained through the FGDs and KIIs evidences the complementarity of local and scientific knowledge, even with the coarse scale

global datasets explored here, implying the potential of blending these to provide effective early warning of flash floods.

## 5.2 Combining local and scientific knowledge toward People-Centred early warning systems

None of the communities we interviewed in Karonga district had access to a formal warning before recent events, neither based on their knowledge of the hydro-meteorological conditions they recognise as possible precursors to flash floods, nor

guidance from SARFFG issued through the DCCMS. Given the knowledge of the communities of the catchments most susceptible to flash floods and the hydro-meteorological conditions that may lead to flash flood events, and that these conditions can be identified in large-scale hydro meteorological datasets such as ERA5, there is clear potential in combining this information into a form of a consensus warning (Plotz et al, 2017) of elevated flash flood risk in the district. Developing warning messages that visualise the indicators that are understood by the recipients of the warnings can contribute to the

credibility of these warnings, helping close the "usability" gap (Vincent et al, 2020), and foster two-way communication between observations of the communities and warning provision (O'Sullivan et al. 2012, Basher 2006). The taking of protective action by recipients of warning messages, if indeed these are received, depends on several factors, including understanding, trust in the provider of warnings, ownership, and personal and contextual relevance (Parker 2009, Moinari and Handmer, 2011, Salit et al 2013). Shah et al, 2012 found that confirmation of warning content through observation of

visual cues in their environment that confirmed the warning content, contributes to the taking of protective actions by recipients. Additional research is needed on how to combine local knowledge and community observations with the scientific forecast information in the provision of warnings to communities such as in Karonga, as careful design warning content and the dissemination and communication methods is required. This design should also acknowledge other signs



communities recognise as precursors to flash flood events which were reported in the FGDs, such as in animal and plant
behaviour. In Karonga, groups of big black Phanga birds flying fast towards the mountains, ant movements and the presence
of butterflies are indigenous signs associated with imminent heavy rains, while frog noises are often heard the night before a
flood. The community of Mwenelupembe village also said they perceived a change in vegetation colour, turning deep green
before the flood. Such careful design is also relevant to avoid diminishing response and trust due to a high false alarm rate as
a result of forecast over-confidence (Morss et al 2016). Though not extensively explored in the FGD we held with
communities, elderly people described an intensification of rain events and a change in meteorological patterns, making the
forecasts based on local knowledge more difficult. This reflects observations made by participants in the research of Sakic
Trogrlic et al. (2019) in the Lower Shire river basin in Malawi, who saw the manifestation of climate change through a
change in rainfall patterns, which negatively influenced the reliability of local indicators. Rising temperatures in Southern
Africa as a consequence of climate change are consistently projected, as are changes to precipitation patterns, though the
projections how precipitation will change is less certain (Engelbrecht, 2015). Further research will be needed to understand if
and how local knowledge will adapt under climate change. Improved forecast information provided by global forecasting
systems, including through integration of local hydro-meteorological observations (Lavers et al, 2019), as well as improved
interpretation of these forecasts at regional level (Jubach and Tokar, 2016) could contribute to reducing false alarm rates.

**Conclusion.**

Using social science-based methods and secondary data sources, we document the knowledge that communities have on the
occurrence and impacts of flash floods in Karonga District in northern Malawi. Thematic analysis of the transcriptions from
focus group discussions with communities, and key informant interviews with local and national experts, revealed which
catchments communities identify as susceptible to flash flooding, as well as the hydro-meteorological signs they recognise as
precursors to flash flood events. This local knowledge was used to guide scientific analysis of the susceptibility of flash
floods in catchments identified as flash flood prone by the communities, as well as the hydro-meteorological conditions prior
to and during documented flash flood events extracted from coarse-scale globally available models and datasets.

The local knowledge of the communities, documented through the focus group discussion and key informant interviews
shows that there is a well-developed knowledge of the occurrence of flash floods in the district, including which catchments
are more susceptible to flash flooding and the factors that may aggravate susceptibility. There is also well-developed
knowledge of the hydro-meteorological conditions they consider as precursors to flash flood events. Interestingly, the
communities identified differences in flash flood events and the impacts these have across the district, including the different
characteristics of flash flood events occurring in the early wet season and in the late wet season. Integrating this local
knowledge with secondary data of flash floods and their impacts in the district contributed to developing a common baseline
of the knowledge and perspectives of the spatial and temporal occurrence of flash floods and their triggers in Northern
Malawi.



Our geo-morphological analysis of the catchments, based on the geometric attributes and indicators extracted from SRTM DEM data, Normalized Difference Vegetation Index (NDVI) from the Copernicus Global Land Service, and national soil and land-cover datasets corroborates the variability of flash flood susceptibility of the catchments in the district described by the communities. Similarly, indicators of the hydro-meteorological conditions and patterns extracted from GSMaP satellite precipitation estimates and ERA5 global reanalysis datasets, validate the precursor signs communities report, including the different characteristics of events across the district and across the wet season. This demonstrates that flash floods happen the way people say, as well as how local knowledge can be used to guide and validate scientific analysis.

Through combining the local knowledge and the scientific analysis of hydro-meteorological conditions and geomorphological patterns, we developed a common understanding of flash floods in Northern Malawi. We identified that the occurrence of flash floods and their impacts differ both spatially and temporally. The analysis suggests that flash floods in the South of Karonga District are mostly triggered by localised convective storms, aggravated by lower vegetation cover, while flash floods in the North of the district are triggered by longer duration orographic rainfall, also extending later in the wet season, with events lasting longer due to the lower infiltration rate.

This common understanding of flash flooding is developed through a bottom-up approach that starts from the risk knowledge and interpretation of communities affected by flash floods and using this to guide the analysis of geomorphological and hydro-meteorological conditions. This holds significant potential in developing a more people-centred early warning of flash floods in those areas of the world where high-resolution forecast data may not be available. Though warnings triggered by indicators extracted from the global datasets used here result in over-confident forecasts, our results highlight the potential these datasets have, even at the local scale. Combining the local and scientific knowledge and understanding, and using commonly understood cues, will lead to better efficiency in triggering action prior to flash flood events, which is crucial to reduce flash flood impacts in vulnerable communities.



**Appendices.**

Appendix A:

**Table A1: List of indicators included in the Principal Component Analysis per categories, resulted indicator loading factors, and category weights (a, b, c and d).**

**a) PC[geom]: Catchment geometry characteristics**

| Variable | Symbol | Loading Factor |
|---|---|---|
| Area | Area | 0.29 |
| Length to width | LtoW | 0.24 |
| Basin Circularity | Cb | 0.13 |
| Time of Concentration | Tc | 0.34 |
| *Weight a* | *0.50* | |

**b) PC[hypsom]: Catchment hypsometry characteristics**

| Variable | Symbol | Loading Factor |
|---|---|---|
| Relative Elevation Ratio | Rel_Relief | 0.38 |
| Elevation Relief Ratio | Elev_RR | 0.24 |
| Mean Slope | Slope | 0.38 |
| *Weight b* | *0.10* | |

**c) PC[drain]: Catchment drainage network characteristics**

| Variable | Symbol | Loading Factor |
|---|---|---|
| Drainage density | Dd | 0.25 |
| Drainage relief ratio | D_RR | 0.50 |
| Basin bifurcation ratio | Rb | 0.25 |
| *Weight c* | *0.20* | |

**d) PC[Surface]: Catchment surface chacacteristics**

| Variable | Symbol | Loading Factor |
|---|---|---|
| Soil Index | Soil | 0.43 |
| NDVI_december | NDVI_Dec | 0.57 |
| *Weight d* | *0.20* | |

**Data availability.** The historical flood impact database of Karonga built for the current research is available on demand. The
dataset is restricted to analysis only, and cannot yet be published openly, as it contains data entries that are property of the
Department of Disaster Management Affair (DODMA). Dataset reference: Bucherie, A.: Karonga historical flood
occurrences and impacts dataset (2000-2018) [Data set]. Zenodo. http://doi.org/10.5281/zenodo.4661438, 2021.

**Supplement link.** Supplementary materials sent for review as .zip file

**Author contribution.** Agathe Bucherie. Conceptualization, Data curation, Formal analysis, Investigation (Field data
collection), Methodology, Writing- Original draft preparation/ Review & Editing. Micha Werner. Conceptualization,
Methodology, Software (providing data extraction code), Supervision, Writing- Original draft preparation/ Review &
Editing. Marc van den Homberg. Conceptualization, Methodology, Funding acquisition, Supervision, Writing- Review &



Editing. Simon Tembo. Project administration (Field Campaign organisation), Investigation (Primary data collection- FGD facilitation), Writing- Review.

**Competing interests.** The authors declare that they have no conflict of interest.

**Acknowledgement.** The field research work was funded by the European Civil Protection and Humanitarian Aid Operations
(ECHO) II Enhancing resilience in Malawi program in Malawi. First of all, we are extremely grateful to the meteorologists Charles Vanya and Victor M. Phiri (from the Malawi Department of Climate Change and Meteorological Services, DCCMS), and Richard Nyoni (from Self Help Africa) for their involvement in the research, confidence in sharing with us valuable data, information and experiences on flash flooding in Northern Malawi. This paper and research would not have been possible without the support of the Malawi Red Cross Society with organizing and conducting the field research in
Karonga district. We sincerely thank all participants of the Focus Group discussions and the Key Informant Interviews for their time and enthusiasm to share their valuable knowledge. We also would like to address a special acknowledgement to our Nyasatime reporter Andrew Mwenelupembe, for his dedicated journalistic support in the setting up, facilitation, translation and transcription of the Focus Group Discussions. Finally, we would like to thank the 510 Global team, and particularly Marijke Panis and Aklilu Teklesadik for the valuable technical support and brainstorming discussions, as well as
Hans van der Kwast, senior lecturer at IHE-Delft, for his QGIS inspiration on the hydrology and geomorphology analysis. This research was largely undertaken by the lead author as a part of her MSc studies at IHE Delft.

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
