# Peer review of "Flash Flood warning in context: combining local knowledge and large-scale hydro-meteorological patterns"

_Natural Hazards and Earth System Sciences, 2021_

## Author Comment (AC2)

**Answer to Reviewer 2**

Thank you very much for the detailed review of our paper and constructive comments. Please find our response to your suggestions, which we have divided into five parts to address each of the comments raised.

*PART 1:* "My main concern has to do with the lack of consistency between the title, the objectives of the research and the results obtained after implementing the proposed methodology. Thus, the combination of local knowledge with larger global scale datasets allows improving the classification of catchments according to their susceptibility to flash floods, but it can hardly be used to generate meaningful and trustworthy warning systems, as it is stated at the end of the manuscript's introduction. Therefore, I propose that, instead of improving early warning, the focus of the research be on improving flash flood susceptibility mapping based on the joint use of local knowledge and global datasets. In the absence of accurate and high spatial resolution data the definition of early warning systems is extraordinarily complex, as accurate and reliable predictions must be provided for a time window not exceeding 6 hours to help improve the capabilities of civil defense and other governmental and non-governmental agencies to enhance preparedness to cope with flash flood disasters and other associated emergencies."

**Answer :** We thank the reviewer for sharing this concern, and can understand the concern well, acknowledging that an important part of our research relates to flash flood susceptibility mapping and risk knowledge. However, our main objective in this paper is to provide a better context for early warning for flash floods. As we outline in the introduction (lines 67-82) we consider early warning within the broader context of the warning process, as proposed by the people-centred multi-hazard early warning framework proposed by the WMO (2015, 2018). This recognises four essential components to early warning; (i) risk knowledge; (ii) monitoring and warning; (iii) communication and dissemination; and, (iv) response capacity. The second of these four steps considers mainly the technical component of the early warning system itself, which we fully agree with the reviewer is very complex, particularly in the context of accurately providing predictions of flash floods. Indeed in the discussion section we present initial findings of using global datasets to inform predictions of flash floods at different levels of spatial aggregation, and show that this is indeed complex and may require higher resolution datasets to improve skill.

Acknowledging the understanding of risk as an essential first step to build effective early warning, we argue that doing so by starting from the risk knowledge and interpretation of risk by communities at which warnings ultimately are targeted, can lead to better efficiency in triggering appropriate response to reducing impacts of flash flood events, and is quintessential to developing more people-centred early warnings. This concept is also very much aligned with research on impact-based forecasting, which emphasises the need for a common understanding of risk shared by both forecast providers (e.g. civil defense and other governmental and non-governmental agencies) and recipients, and the need for warning message content to align with the (personal) interpretation of risk

recipients have, including environmental cues (Basher, 2006; Martin, 2012; Vincent et al 2020; Šakić Trogrlić, et al., 2019 - note all references already included in the main paper).

Within the context of first building a common understanding of flash flood risk and thus contributing to improved early warning, our approach combines local knowledge with large-scale contemporary scientific datasets. We underline that flash flood risk encompasses more than susceptibility. It includes the commonly considered risk dimensions of hazard and exposure, vulnerability and coping capacity. We specifically included the broader risk context, so not only hazard and exposure (which directly relates to susceptibility) but also vulnerability of communities affected. For example, our selection of catchments to focus on was based on the impact communities experienced.

We agree that our research does not aim at improving flash flood warning with more precise and higher lead time forecasts. Rather, our approach is that, in the absence of accurate and high spatial resolution data, warnings could be improved in the broader context through integrating local knowledge and lower resolution globally available scientific knowledge on the factors leading to flash flooding. Despite forecasts of flash floods established using the identified indicators being over-confident as discussed, our results evidence the potential of linking larger scale global datasets with local knowledge, using commonly understood cues, to provide meaningful early warning to local communities, and help triggering effective emergency response to reduce the impacts of flash flood events in vulnerable communities.

In addition, in the discussion section we address how local knowledge can be integrated into flash flood triggers, give recommendations at which scale to develop future warning, and contribute to improved relevance of large scale scientific data to local needs and decision making by understanding the social and behavioural factors that govern user actions.

To clarify what we imply in our hypothesis in generating meaningful and trustworthy warning, we propose to be more explicit in this in the last paragraph of the Introduction (lines 83-94). The text below shows the amended paragraph, with changed text in bold.

*In this paper we explore local knowledge and science-based information on the occurrence of flash floods in rural communities in Karonga District in northern Malawi. We hypothesize that local knowledge can complement the information contained in larger scale global datasets, and that the combination of local and scientific knowledge can contribute to the development of meaningful and trustworthy early warning, **within the context of the people-centred early warning framework (WMO 2015, 2018), which recognises that effective early warning builds on four key interrelated elements; (i) risk knowledge; (ii) monitoring and warning; (iii) dissemination and communication; and, (iv) response capacity.** North Malawi is an example of an area with high flash flood risk where the population is extremely vulnerable due to low coping capacities. Through interviews with impacted communities, we develop **a shared knowledge of risk** through a joint understanding of the root causes of flash floods in the area, the impacts these have and where these are more likely to occur. In the interviews*

*we also consider the local knowledge of meteorological and hydrological signs communities recognize as precursors to flash flood events. We then interrogate available information on catchment geomorphology and hydro-meteorological conditions derived from large-scale global models and satellite datasets to examine if these provide useful information congruent to that local knowledge. Our aim is to reconcile these scientific data with local knowledge of flash floods to inform the implementation of people-centred flash flood warnings and foster the taking of early action by communities.*

Additionally we propose to extend the sentence in line 443 of the discussion where we discuss the skill of the coarse global datasets in providing forecasts that could trigger flash flood warnings. We further emphasise the point raised by the reviewer that this is very complex, particularly in the absence of high resolution datasets.

Proposed changes to lines 443-444:

**These results underscore the complexity of predicting the occurrence of flash floods to trigger warnings at the local scale, particularly using coarser global datasets in absence of the availability of high-resolution observational data and numerical weather predictions. Despite this difficulty of providing predictions at the very local scale, the results do highlight the potential these datasets have in providing guidance on the potential occurrence of flash floods in the district.**

*PART 2: On the other hand, some parts of the manuscript deserve to be explained in more detail. Thus, in section 3.1, Building the common knowledge of flash floods, the methodological approach used to identify the actors needs to be explained in detail. The number of actors selected, and their characteristics should also be explained thoroughly.*

**Answer :** A more detailed methodology to select the actors for the Key Informant Interviews (KII), as well as the participants of the Focus Group Discussions (FGD) is provided in the Supplementary material 3.1. The list of actors targeted for the KII at national and Karonga district levels, together with their roles is described in section 3 and 4. Our objective was to interview representatives from a wide range of institutions, including from disaster, meteorological and hydrological governmental institutions, universities, and disaster practitioners from locally active NGO. The final list of actors presented in the paper was the result from availability and in-country contacts. We ensured that all KII and FGD were prepared to comply with the COREQ qualitative research criterias (Tong et al., 2007). We will add more information in the main paper on the method used to identify actors. This explanation will be added in line 118.

*PART 3 : Furthermore, I consider that the description of the results on the building of knowledge on flash floods needs to be expanded. Thus, in the explanation of the methodology, it is stated that the construction of such knowledge was based on interviews with national and local key actors, as well as on the holding of focus groups at the community level. However, the results provided are of a general nature and are therefore not in line with the methodology.*

**Answer :**

The results of the building of knowledge on flash floods are indeed partly extracted from the Key Informant Interviews and Focus Group Discussions, articulated around similar questionnaires patterns, conducted during the field data collection (primary data). As described in the methodology section (line 138), "we complement these primary data with secondary data on historical flash flood events, their location, date and recorded impacts". The results from the primary and secondary data were extracted and analysed around 5 themes: (i) Flash flood definition; (ii) occurrence of flash flood events; (iii) impacts of flash floods; (iv) risk perception, and (v) the signs leading to flash floods. These are the 5 themes developed in the result section, and summarize for the first time the knowledge of flash floods in Karonga region. Only the results of themes ii) and iii) come from the integration from primary and secondary data analysis.

Our research is interdisciplinary, as it is "based upon a conceptual model that links or integrates theoretical frameworks from those [different] disciplines, uses study design and methodology that is not limited to any one field, and requires the use of perspectives and skills of the involved disciplines throughout multiple phases of research process." (Aboelela et al., 2007). We combine quantitative approaches as common in the hydro-meteorological/physical sciences and qualitative approaches as common in the social sciences. The qualitative approach was the most appropriate to study local knowledge as local knowledge is produced and used by people (Šakić Trogrlić, et al., 2020). Bryman (2012) describes qualitative research as: i) providing meaning through the eyes of the individuals studied, ii) putting emphasis on context, iii) providing process perspectives on the phenomena studied, iv) being flexible, and v) being grounded in data.

This may explain why the results of the qualitative research provided are of a more "general" or descriptive nature (although with local specificity, representing the climate science of the area (Nkuba et al., 2021) which is in line with the social sciences methodology used for understanding local knowledge. Based on your suggestion, we will expand on the qualitative process used for this research in the method section (3.1) of the manuscript. In addition, We will briefly reiterate the method before presenting the results section (4.1), and rephrase the results section to be more specific and clarify how the knowledge that is reported is built on the FGD and KII through extracting quotes from the interview transcripts along the five specified themes. We will also clarify where the risk knowledge that is established based on the FGD and KII (primary data) is supported through the secondary data.

PART4 : In section 2, study area

- The geomorphological, land use and soil type characteristics of the study area must be explained in more detail.
- A description of a significant flash flood event which has taken place in the study area would also help to contextualize the research undertaken.
- In this section, I also believe that the vulnerability of the population exposed to flash floods needs to be explained.

**Answer :** Thank you for the comments on section 2 where we describe the study area, which is indeed somewhat brief. We agree that it is relevant to provide more context to the reader and this will be included in the revised manuscript. We will add the following information in the study area section with background information about Karonga's surface characteristics, population vulnerability, and an example from secondary data describing a major flash flood event and associated impacts.

*"Characterised by strong erosion of the crystalline baserock, filling the rift valley with quaternary sediments, Karonga soil types are mostly sandy. Land use along the flat lake shore is almost entirely cropland (mostly rice, maize and cassava) while the hills and plateau are covered by bushes and open forest. The population of Karonga is rural, and is distributed mainly on the flat lake shore in small communities. The economy depends upon subsistence agriculture and fishing. With a poverty incidence of 57.1 % (IFPRI, 2019), Karonga district is poorer than the average of Malawi.*

*Flood events recur annually and are particularly damaging in Karonga due to poor infrastructure, growing population and increase of farming in flood risk areas, and difficulties for the population to get warning due to lack of access to communications.*
*Large scale damaging events can occur in the district of Karonga, like the floods of 12-16 April 2018, affecting 4069 people, destroying 433 houses, and killing 4, as reported by the Department of Disaster Management Affairs of Malawi (DoDMA). Smaller scale and isolated flash floods have also impacted the district, like the event of the 1st february 2018 affecting 1175 persons and damaging 42 houses and 397 hectare of crops in the small village of Mwenelupembe according to DoDMA."*

*https://floodlist.com/africa/malawi-floods-northern-region-january-2018*
*https://www.nyasatimes.com/one-person-dies-karonga-floods-mp-mwenifumbo-fears-damage/*

PART 5:  OTHER COMMENT

In Figure 1, what does the colour scale in the figure on the left mean? Please clarify in the legend or in the figure caption.

**Answer :** The flood recurrence map and definition (Low, Medium, High) was defined by DoDMA (Department of Disasters Management Affairs - VAM - MEPED) for the Integrated Context Analysis report of  2014, based on the historical number of flood events in the period 2000-2013. We will add a brief explanation to the figure caption.

The layer is available at the following link:
https://geonode.wfp.org/wfpdocs/malawi-frequency-of-flood-events-between-2000-and-2013-june-2014/

ADDITIONAL REFERENCE USED IN THE ANSWER:

Aboelela, S.W., Larson, E., Bakken, S., Carrasquillo, O., Formicola, A., Glied, S.A., Haas, J. and Gebbie, K.M. (2007), Defining Interdisciplinary Research: Conclusions from a Critical Review of the Literature. Health Services Research, 42: 329-346.

Bryman, A., 2012. Social Research Methods. 4th ed. New York: Oxford University Press.

IFPRI. 2019. IFPRI Key Facts Series : Poverty May 2019 Background to the Integrated Household Surveys (IHS)

Sakic Trogrlic, R. (2020). The role of local knowledge in community-based flood risk management in Malawi (Doctoral dissertation, Heriot-Watt University).

Nkuba, M. R., Chanda, R., Mmopelwa, G., Mangheni, M. N., Lesolle, D., Adedoyin, A., & Mujuni, G. (2021). Determinants of pastoralists' use of indigenous knowledge and scientific forecasts in Rwenzori region, Western Uganda. Climate Services, 23, 100242.